# Excellent Toughening of 2,6-Diaminopyridine Derived Poly (Urethane Urea) via Dynamic Cross-Linkages and Interfering with Hydrogen Bonding of Urea Groups from Partially Coordinated Ligands

**DOI:** 10.3390/polym11081320

**Published:** 2019-08-07

**Authors:** Ailing Sun, Wenjuan Guo, Jinping Zhang, Wenjuan Li, Xin Liu, Hao Zhu, Yuhan Li, Liuhe Wei

**Affiliations:** 1College of Chemistry and Molecular Engineering, Zhengzhou University, Zhengzhou 450001, China; 2Zhengzhou Key Laboratory of Elastic Sealing Materials, Zhengzhou 450001, China

**Keywords:** toughening, strengthening, dynamic cross-linkages, metal-ligand, TPU

## Abstract

Conventional approaches to synthesize thermoplastic polyurethane (TPU) with excellent robustness are limited by a competing relationship between soft and hard segments for tuning mechanical properties in terms of chain flexibility and micro-phase separation. Herein, we present a facile and effective way of simultaneously improving the tensile strength, elongation, and toughness by constructing dynamic cross-linkages from metal-ligand interaction between Zn^2+^ and pyridine moiety in backbone of poly(urethane urea) (PUU) derived from 2,6-diaminopyridine and poly(propylene glycol). It was found that a Zn^2+^/pyridine ratio of 1:4 is the most effective for improving robustness. Specifically, tensile strength, elongation, and toughness could be remarkably increased to 16.0 MPa, 1286%, and 89.3 MJ/m^3^ with 226%, 29%, and 185% increments compared to uncomplexed PUU, respectively. Results from UV-vis, Fourier transform infrared spectroscopy (FTIR), cyclic tensile tests, and stress relaxation reveal that metal-ligand interaction significantly interferes with the hydrogen bonding of urea groups, thus leading to weakening of stiffness. Furthermore, half of vacant ligands enable dynamic complexation during stretching, which consequently ensures constant noncovalent cross-linkages for constraining mutual chain sliding, contributing to simultaneous improvement of tensile strength, elongation, and toughness. This work provides a promising approach for designing TPU with excellent robustness.

## 1. Introduction

Thermoplastic polyurethane (TPU) is widely used in daily life and industrial fields due to its exceptional processability and mechanical properties amid rubbers and plastics. It principally consists of a long-chain polyester or polyether with relative high molecular weight as soft segments, short-chain extenders and urethane and/or urea groups as hard segments. Conventionally, tuning the mechanical properties, thermal resistance, and other special features of TPU lies in altering the structure of the soft and hard segments, because its elasticity is essentially originated from the physical network constructed by micro-domains derived from hydrogen bonded hard segments [1,2,3,4].To date, it is still a great challenge to design perfect structure and condensed state for improving tensile strength, elongation, and toughness at the same time, since soft and hard segments have mutually competitive dependence on chain mobility. 

Many efforts have been devoted to strengthening and toughening TPU via incorporation of fillers [5,6,7,8,9] and in-situ synthetic methods [10]. Synthesizing TPU with high performance refers to molecular structure and condensed state design that consequentially controls network, soft and hard segments. Recently, sacrificial bonds, defined as covalent or noncovalent bonds which perform rupture of their binding prior to the cleavage of major structural linkage upon applied force [11,12], are prevailing in designing materials with self-healing ability and enhanced mechanical properties. Hydrogen bonds in TPU, induced by either urethane or urea groups, could actually act as sacrificial bonds; however, the weak binding energy somehow limits the enhancing ability. For instance, the presence of numerous hydrogen bonds in TPU generates improved stiffness and strength, and decreased elongation, while a low amount of hydrogen bonds is not favorable for improving strength. Metal-ligand interaction is basically stronger in binding energy than hydrogen bonding [13,14], and thus some researchers have incorporated it into elastomers for constructing dynamic noncovalent network in order to improve both strength and toughness or confer self-healing ability [15,16,17,18,19,20,21]. These approaches arranged ligands as side groups of polyolefin elastomers or thermosetting polyurethane (PU) elastomers at chain ends. In these cases, the metal-ligand interaction either imposes a negligible effect on backbone mobility or is accompanied by the effect of the physical or chemical network. Some researchers have recently developed TPU with ligands aligning in backbone, which showed effective improvement of mechanical properties. For instance, Hong et al. [22] and Zhang et al. [23] utilized triazol-pyridine derivative and curcumin as both chain-extender and ligand. The former authors found that complete complexation of ligands resulted in optimal mechanical properties, but, unfortunately, less interpretation of structure-properties was mentioned. The latter also found that the robustness of TPU could be effectively enhanced with a specific metal/ligand ratio, but the influence of metal-ligand interaction on structure merely referred to distance of hard domains. Therefore, how the metal/ligand ratio and coordinative bonds affect hard segments and formation of noncovalent cross-linkages still requires investigation. 

Herein, we present a routine approach to synthesize poly(propylene glycol)-based poly(urethane urea) (PUU) by using 2,6-diaminopyridine as chain extender and ligand. Poly(propylene glycol) is selected as a soft segment due to its noncrystallinity and high chain flexibility. 2,6-Diaminopyridine is unique because it could serve as a backbone ligand and its adjacent urea groups might be notably affected by coordinative bonds. Although a recent paper reported that 2,6-diaminopyridine and poly(tetramethylene ether glycol)-based TPU showed improved robustness and self-healing ability [24], our work focuses on revealing the role of Zn^2+^/pyridine ratio and its metal-ligand interaction in affecting hard segments and formation of noncovalent cross-linkages. It was found that the optimal Zn^2+^/pyridine ratio for effectively improving robustness does not necessarily belong to complexation of entire ligands, and taking advantages of interfering with hard segments by metal-ligand interaction is the key for tuning strength, elongation, and toughness at the same time.

## 2. Materials and Methods

### 2.1. Materials and Synthesis

Poly(propylene glycol) (PPG) (Mn = 2000 g/mol) was purchased from Bluestar (Zibo, China) and used without further purification. 4,4’-Diphenylmethane diisocyanate (MDI) was purchased from Wanhua (Yantai, China) and used without further purification. 2,6-Diaminopyridine, 1,6-hexanediamine (HMD), and zinc trifluoromethanesulfonate (Zn(OTf)_2_) were purchased from Adamas (Shanghai, China) and used without further purification. *p*-Chlorophenol and dibutyltin dilaurate (DBTDL) were purchased from Aladdin (Shanghai, China) and used without further purification. *N*,*N*-dimethylformamide (DMF), acetonitrile, and tetrahydrofuran (THF) were purchased from Kermel (Tianjin, China) and used after redistillation for removing residual moisture.

Typical synthesis of PPG and 2,6-diaminopyridine-based poly(urethane urea) (PUU-Py) is presented in Scheme 1. PPG (12.00 g, 6 mmol) in a dried glass vessel equipped with a mercury thermometer and a mechanical agitation was heated to 120 °C under vacuum for 40 min to remove residual moisture in raw material, and then cooled down to 80°C. MDI (3.153 g, 12.6 mmol) was added into glass container for 1 h at 80 °C. DBTDL (0.7584 mg, 0.00125 mmol) was then added into glass container for another 1 h, resulting in pre-polymer. Then 2,6-diaminopyridine (0.6842 g, 6.27 mmol) as chain extender dissolved in anhydrous DMF (10 mL) was added into the container for 1 h. *p*-Chlorophenol (0.0848 g, 0.66 mmol) was added into the container as blocking agent. The reaction was continued until the isocyanate group (NCO) peak disappeared from the Fourier transform infrared spectroscopy (FTIR) spectrum, which required 30 min. Zn^2+^/pyridine ratio is defined as molar ratio of Zn^2+^ to pyridine moiety. To introduce a ratio of 1:4, Zn(OTf)_2_ (0.5698 g, 1.568 mmol) dissolved in acetonitrile (10 mL) was added into the container for 10 min to ensure homogeneous mixture, and the as-prepared elastomer is denoted as PUU-Py1/4. PUU-Py1/6, PUU-Py1/3 and PUU-Py0/1 elastomers with various Zn^2+^/pyridine ratios were prepared by an identical method with adjusting content of Zn(OTf)_2_. Synthesis of PUU-HMD was slightly different, i.e., the pre-polymer was preliminarily end-capped with *p*-chlorophenol prior to chain-extending with HMD. All reaction procedures were carried out in argon atmosphere and mechanical agitation.

### 2.2. Methods

#### 2.2.1. Preparation of Films

As-synthesized product was poured into a rectangle Teflon mold with dimensions of 120 mm × 90 mm × 15 mm. The mold was placed in an oven and heated at 80 °C for 48 h, then put the mold into vacuum, drying at 80 °C for 24 h to remove residual solvent.

#### 2.2.2. Mechanical Properties Tests

Dumbbell specimens were cut for tensile strength tests and cyclic tensile tests, which were both performed using a tensile tester (TOPHUNG, TH-8203A, Suzhou, China) loaded with a 500 N load cell. The tensile strength tests were conducted at a constant speed of 100 mm/min at room temperature. The cyclic tensile properties were measured by stretching specimens to 400% strain at a constant speed of 50 mm/min. Two cycles were carried at an interval of 0 min.

#### 2.2.3. Stress-Relaxation Tests

The test specimens were bar-shaped films with approximate dimensions of 15 mm × 6 mm × 1.0 mm. The stress-relaxation was tested by TA Q800 Instruments (TA instruments, New Castle, DE, USA). Samples were stretched to 100% and the constant strain amplitude was maintained to measure the relaxation of the stress for 60 min.

#### 2.2.4. Fourier Transform Infrared Spectroscopy (FTIR)

Bulk samples were brought to perform Fourier transform infrared spectroscopy (FTIR) in attenuated total reflectance (ATR) mode. Solution samples of various PUU-Py elastomers in DMF (2 g/mL) were brought to carry out FTIR in transmitted mode at 25 °C from Bruker (VERTEX 70v, Karlsrule, Baden-Wurttemberg, Germany).

#### 2.2.5. UV-vis Spectroscopy

UV-vis spectroscopy was carried out using Hewlett Packard 8453 UV-vis Spectrophotometer (G1103A), where 17 μL of Zn(OTf)_2_ solution in methanol was each time added to 2 mL of PUU-Py solution in THF (0.25 mg/mL). UV-vis spectroscopy of PUU-HMD was carried out by its solution in DMF because of relative good solubility compared to in THF.

## 3. Results and Discussion

Recent literature reported [15,16,20,25] that dangling ligand moieties along the backbone was an effective means of enhancing mechanical properties. Our strategy to prepare robust and durable PUU elastomer lies in the incorporation of ligand moiety into backbone in order to form metal-ligand interaction in backbone. Another important factor is that the pyridine moiety might impose interference with hydrogen bonding of the adjacent urea groups, which would compromise micro-phase separation of hard segments and sequentially decrease stiffness and confer large elongation.

### 3.1. Mechanical Properties

Tensile tests were performed to reflect effect of metal-ligand interaction on robustness of as-prepared elastomers. As shown in Figure 1a, tensile strength of PUU-Py elastomers underwent drastic changes upon introduction of Zn^2+^. Comparison of tensile strength and toughness is shown in Figure 1b,c. Specifically, tensile strength, elongation, and toughness of PUU-Py0/1 was measured to be 4.9 MPa, 995%, and 31.3 MJ/m^3^, respectively. By contrast, PUU-Py1/2, PUU-Py1/3, and PUU-Py1/6 have apparent improved robustness, because the individual tensile strength increased to 6.5 MPa, 8.8 MPa, and 6.7 MPa with paralleled elongation while the corresponding toughness was calculated to be 33.1 MJ/m^3^, 50.5 MJ/m^3^, and 47.5 MJ/m^3^, respectively. It is surprising that PUU-Py1/4 has the highest tensile strength, elongation, and toughness, i.e., 16.0 MPa, 1286%, and 89.3 MJ/m^3^ with 226%, 29%, and 185% increments in comparison with PUU-Py0/1. 

Introduction of a chemically covalent network of TPU usually favors in improving merely tensile strength, but is quite difficult to improve tensile strength and elongation at the same time because the extensibility of curled chains is severely constrained by covalent cross-linkages [26]. Analogous to chemical network, physical network may also restrict complete extension of curled chains. To elucidate it clearly, curled chains of soft segment may undergo thorough extension or even chain cleavage, before large hard domains perform complete deformation and chain extension. Therefore, tuning hard domains plays a pivotal role in sufficient exertion of strengthening and toughening.

In the present work, incorporation of Zn^2+^ seems to successfully induce coordinative bonds with backbone pyridine moieties, because the mechanical properties were remarkably influenced. Such metal-ligand interaction also indicates that hard domains were compromised due to increased elongation at break for PUU-Py elastomers complexed with Zn^2+^. It can be observed that PUU-Py1/4 exhibited an upturn of slope around 800% instead of fracture. This result indicates that dislocation of chains was restricted when curled chains were forced to stretch to the largest extent. This phenomenon implies that Zn^2+^ played a critical role in influencing mechanical properties and the optimal ratio for enhancing robustness did not necessarily correspond to high ratio of Zn^2+^/pyridine. Considering that Zn^2+^/pyridine ratio of 1:4 resulted in the most exceptional mechanical properties, it is reasonable to assert that metal-ligand interaction successfully induced noncovalent cross-linkages in our system.

### 3.2. FTIR Spectroscopic Analysis

FTIR was performed to validate metal-ligand interaction, as displayed in Figure 2. In the case of PUU-Py0/1, 1601 cm^−1^ and 1727 cm^−1^ was individually assigned to aromatic skeleton vibration and stretching of urethane carbonyl (C=O) (Figure 2a), while the bending of free pyridine moiety at 1590 cm^−1^ and 1572 cm^−1^ was camouflaged by aromatic skeleton vibration compared to neat 2,6-diaminopyridine (Figure 2b). The peak at 1540 cm^−1^ was assigned to be amide II, i.e., combination of N–H bending and C–N stretching [27]. Peaks at 1694 cm^−1^ and 1645 cm^−1^ were stretching of free and ordered hydrogen-bonded urea carbonyl (C=O) [4,27,28,29,30,31]. By contrast, a new peak emerging at 1621 cm^−1^ was assigned to the shifted bending of pyridine moiety due to its coordination with Zn^2+^ [32]. Moreover, an explicit peak could be observed at 1672 cm^−1^, accompanied by disappearance of peaks at 1694 cm^−1^ and 1645 cm^−1^. This is also indicative of metal-ligand interaction because both free and ordered hydrogen-bonded urea carbonyl groups adjacent to pyridine moieties were significantly affected [27,31,33,34]. Interestingly, urethane carbonyl at 1727 cm^−1^ remained unchanged, indicating that Zn^2+^ ions did not coordinate with urethane groups. Figure 2c presents FTIR spectra of PUU-Py elastomers in solvent of DMF. Peaks at 1694 cm^−1^, 1658 cm^−1^ and 1633 cm^−1^ were ascribed to free, disordered and ordered hydrogen-bonded urea carbonyl [4,27,31]. Since PUU-Py chains in solution were free to perform chain motion, hydrogen bonding could be easily fulfilled so that the peak of free urea carbonyl was the weakest, the disordered hydrogen-bonded was the strongest, and the ordered hydrogen-bonded was medium compared to that of bulk samples. It is observable that the relative intensity of peak at 1633 cm^−1^ weakened as the content of Zn^2+^ increased, which also confirms the interference of hydrogen bonding by metal-ligand interaction.

### 3.3. UV-vis Spectroscopic Analysis

Coordination behavior of ligand in PUU-Py with Zn^2+^ was investigated by UV-vis spectra. As shown in Figure 3a, the feature peak of free pyridine moiety at λ = 308 nm diminished as Zn^2+^/pyridine ratio increased, while a new peak emerged at λ = 315 nm, which was associated with ligand-to-metal charge transfer [35,36]. The variation of the two featured peaks versus the Zn^2+^/pyridine ratio illustrated in Figure 3b reveals that complete complexation required a Zn^2+^/pyridine ratio of ~0.5, because the absorbance gap between λ = 308 and 315 nm reached to a nearly constant value and further increasing ratio merely resulted in synchronous change of absorbance. These UV-vis spectra results indicate that coordination took place through pyridine-N atom and Zn^2+^. Furthermore, the adjacent amide-N atom and oxygen atom of urea carbonyl might have participated in coordination. 

Absorbance of urea and urethane carbonyl at λ = 290 nm underwent a decrease, instead of shift, as the Zn^2+^/pyridine ratio increased (Figure 3a,b), indicating that the carbonyl groups were affected by coordination. PUU-HMD, a control to PUU-Py using 1,6-hexanediamine as chain-extender, also displayed a decrease of absorbance at feature peaks of urea and urethane carbonyl (λ = 290 nm) and aromatic ring (λ = 269 nm) as shown in Figure 4. The former implies urea and urethane carbonyl probably took part in coordination, which was analogous to coordination of curcumin with Eu^3+^ [23]. The latter was assigned to be B band of aromatic ring of MDI, which could have been affected in absorbance by its possibly coordinated neighboring urea-N atom. With reference to the work of Li et al. that metal ions could coordinate with either oxygen atom or nitrogen atom of amide [37], the coordination configuration remains unclear from UV-vis spectra as shown in Figure 3 and Figure 4. However, with regard to FTIR spectra shown in Figure 2a, such uncertainty can be solved. Firstly, stretching of the urethane carbonyl peak at 1727 cm^−1^ and the combination of N–H bending and C–N stretching at 1540 cm^−1^ (amide II) exhibited no shift upon introduction of Zn^2+^. Secondly, both free and ordered hydrogen-bonded urea carbonyl underwent remarkable shifts. This result explicitly indicates that only the oxygen atom of the urea carbonyl participated in coordination with Zn^2+^ according to related analysis of the literature [32]. Thus, the coordination configuration could be illustrated as in Scheme 2.

### 3.4. Analysis of Cyclic Tensile Tests

Cyclic tensile tests were performed to further reveal how metal-ligand interaction affects mechanical properties. Figure 5a exhibits curves of the first loading-unloading cycle to compare hysteresis loop. PUU-Py0/1 has apparently the largest area of hysteresis loop (mechanical hysteresis) than the other elastomers. Mechanical hysteresis related to delayed recovery of chain configuration and conformation caused by inter-chain friction. Hydrogen bonding induced by the urea and urethane groups was the major inter-chain friction that constrained re-configuring and re-conforming chains for PUU-Py0/1. For PUU-Py elastomers incorporated with Zn^2+^, coordinative bonds significantly interfered with the hydrogen bonding of urea, as validated by FTIR in Figure 2, thus recovery of the chain configuration and conformation was favored, generating a smaller hysteresis loop. Once Zn^2+^ was introduced into PUU-Py, hysteresis energy sharply decreased to a relatively steady level, regardless of the Zn^2+^/pyridine ratio, as shown in Figure 5b. In the meantime, comparison of residual strain also reflected metal-ligand interaction affecting hydrogen bonding. Firstly, re-formation of coordinative bonds happened in a slower way than hydrogen bonding, according to principle of ‘strong means slow’ [13]. Secondly, metal-ligand interaction compromised total amount of hydrogen bonds. Hence, the residual strain increased with the introduction of more Zn^2+^ ions (Figure 5b). Therefore, this result confirms coordinative bonds, as shown in Scheme 2, in backbone could impose great intervention upon hydrogen bonding. If urea groups locate far away from pyridine moieties, its hydrogen bonding might take place without intervention, then mechanical hysteresis and residual strain would not be notably affected. 

### 3.5. Analysis of Stress-Strain Behavior

It seems that hydrogen bonds and coordinative bonds have mutually competing roles in tuning mechanical properties, because they actually have great influence on formation of hard domains. The hydrogen bonded hard segments remained intact, so that micro-phase separation resulted in larger hard domains, while coordinative bonds in PUU-Py1/4 notably interfered with hydrogen bonding of urea groups, which reduced the size of hard domains. Thus, PUU-Py0/1 exhibited higher Young’s modulus than PUU-Py1/4, as shown in Figure 6a. The stress-strain curve of PUU-Py1/4 is unique with regard to its large elongation and obvious upturn of slope at ~800%. It can be divided into three parts, as shown in Figure 6b: Regime I occurs at small strain (<~55%); Regime II contains a constant slope in a wide range of strain from ~55% to ~540%; and Regime III has a greater constant slope in the range of strain from ~1064% to 1286%. Regime I includes elastic deformation and destruction of hard domains. Regime II presents a linear relationship between stress and strain in a wide strain range, which means the force for extending curled chain and cleaving hydrogen bonds increases in a balanced way. As strain exceeds ~540%, it has to not only cleave hydrogen bonds but has also enforced chain disentanglement. An extrapolational crossing point of slope at 836% is a symbol that coordinative bonds, i.e., noncovalent cross-linkages, start to dissociate. Due to higher binding energy of metal-ligand interaction than hydrogen bonding [13,14], the stress inducing deformation in Regime III remarkably increases with a greater slope. In other words, coordinative bonds significantly restrict extended chains from undergoing mutual sliding. It is worthy of noting that rupture and re-formation of hydrogen bonds and coordinative bonds probably coexist in Regime III because both of them have the capacity of dynamic binding.

### 3.6. Analysis of Stress Relaxation

Stress relaxation investigation was performed to unveil roles of hydrogen bonding and metal-ligand interaction, as shown in Figure 7. Referring to Figure 1a and Figure 6b, 100% of strain induced destruction of hard domains by cleaving hydrogen bonding and extending of curled chains of soft segments. Obviously, it needs larger external force to induce deformation and destruction of hard domains than to extend curled chains of soft segments. For PUU-HMD, an extremely slow relaxation process was observed, which indicates that the extending of curled chains of soft segments was insignificant, while hard domains gave rise to elastic deformation as well as destruction. This could be explained by the relatively larger size of the hard domains, because the flexible chain-extender favors in micro-phase separation. For PUU-Py0/1, the pyridine moiety had higher steric hindrance than HMD, so that it had smaller sizes of hard domains. Consequently, the extending of curled chains of soft segments and disorderedly-packed hard segments bore more external force. Once the applied force was removed, recovery of configuration and conformation happened faster than PUU-HMD, especially for soft segments. With incorporating Zn^2+^ into PUU-Py1/6, PUU-Py1/4, PUU-Py1/3, and PUU-Py1/2, the size of the hard domains was further reduced due to the intervention of hydrogen bonding of urea by adjacent metal-ligand interaction, and then curled chains of soft segments and disorderedly-packed hard segments dominantly bore the applied force. Therefore, even though reformation of coordinative bonds might take a longer time [13], recovering configuration and conformation could implement in a short time; for instance, the applied force releases 51%, 57%, 66%, and 76% for PUU-Py1/6, PUU-Py1/4, PUU-Py1/3, and PUU-Py1/2 at 20 min, respectively.

### 3.7. Mechanism of Toughening

In order to better understand the excellent engineering effect on PUU-Py elastomers, we propose a strengthening and toughening mechanism from the perspective of metal-ligand interaction. Pristine PUU-Py elastomer contains urea doublet hydrogen bonding and urethane singlet hydrogen bonding, and both induce micro-phase separation, leading to closely-packed hard segment domains that act as physical cross-linking junctions to improve tensile strength and robustness [2,4,27]. PUU-Py elastomers with Zn^2+^ introduced display improved mechanical properties, since the binding energy of metal-ligand interaction is several orders higher than that of hydrogen bonding [14,36]. Coordinative bonds not only interfere with urea doublet hydrogen bonding, validated by FITR shift of urea carbonyl (Figure 2), but also form noncovalent cross-linkages between neighboring chains. This is different from metal-ligand interaction that arranges dangling ligand groups along the backbone, because the rigid pyridine moiety in the backbone reduces chain flexibility [18,38]. 

A molar ratio of 1:6 is not enough to enable adequate pyridine moieties to participate in complexation, so the density of noncovalent cross-linkages is too low, resulting in limited improvement of tensile strength and toughness (Figure 1). PUU-Py1/4 has the best excellent tensile strength and toughness, while PUU-Py1/3 and PUU-Py1/2, Zn^2+^/pyridine ratio exceeding 1:4 does not yield further enhancement of the mechanical properties any more. This result demonstrates that the optimal ratio does not necessarily correspond to the complexation of every ligand, which implies that the presence of vacant ligands facilitates the reformation of noncovalent cross-linkages during stretching and is the key for improving robustness. In the case of pristine PUU-Py, factors that constrain dislocation of chains are mainly chain entanglement and hydrogen bonds induced by the urea and urethane groups. It is easy for external forces to destruct hard segments derived from hydrogen bonding interactions, due to their relatively weak binding energy, accompanied by chain extending, disentanglement, and cleavage in sequence. In the situation of PUU-Py1/4, half of the ligands coordinate with Zn^2+^, generating coordinative bonds between neighboring chains, serving as dynamic cross-linkages (Scheme 3). By contrast, an overdose of Zn^2+^ leaves fewer or no vacant pyridine moieties for re-complexation. These noncovalent cross-linkages impose significant restriction on segmental chain motion and dislocation. The curled chains between cross-linkages are able to stretch to an extreme extent upon external force, before complete rupture of coordinative bonds. Upon rupture of coordination, Zn^2+^ is able to rebind with the remaining half of the vacant ligands. Therefore, tensile strength and elongation at break could be significantly improved, simultaneously, resulting in excellent durability and robustness.

## 4. Conclusions

In summary, the incorporation of pyridine moieties into the backbone of polypropylene glycol-based PUU is able to remarkably improve the robustness of this material, upon being complexed with 1:4 equivalents of Zn^2+^. Specifically, tensile strength, elongation, and toughness of PUU-Py with a Zn^2+^/pyridine ratio of 1:4 could be simultaneously increased to 16.0 MPa, 1286%, and 89.3 MJ/m^3^ with 226%, 29%, and 185% increments in comparison with uncomplexed PUU-Py, respectively. The nitrogen of pyridine moiety and adjacent oxygen of urea participates, in coordination with Zn^2+^, results in dynamic noncovalent cross-linkages. Coordinative bonds impose intervention upon urea hydrogen bonding to compromise micro-phase separation of hard segments, and the mutual sliding of extended chains could be restricted in the process of stretching. Coordinating half of the ligands is found to provide the most effective robustness enhancement, because the remaining half of the ligands are vacant for re-complexation during stretching. This work would provide an effective way of engineering TPU elastomer through molecular design and constructing metal-ligand interaction, which is fundamentally meaningful for industrial applications.

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
