# Peer review of "Excellent Toughening of 2,6-Diaminopyridine Derived Poly (Urethane Urea) via Dynamic Cross-Linkages and Interfering with Hydrogen Bonding of Urea Groups from Partially Coordinated Ligands"

_polymers, 2019, doi:10.3390/polym11081320_

Round 1
Reviewer 1 Report
In this report, authors have synthesized 2,6-diaminopyridine-derived polyurethane that could be dynamically crosslinked with metal ions. By careful manipulation of metal ions (Zn2+) to the pyridine, authors have identified the "sweet spot" with Zn2+/pyridine ratio of 1/4, where there is significant improvement of mechanical properties (tensile strength, elongation and toughness). This improvement is rationalized by the dynamic complexation that not only contributes to a net cross-linkages but also allows for chain sliding. Overall, this work has been conducted in a systematic fashion, data presented well and the finding from this work will impact the design of other class of polymeric materials. I recommend this work to be published.
Author Response
Dear reviewer;
We thank you for your careful consideration and thoughtful comments on our manuscript (ID: polymers-561361).
Best wishes!
Yours sincerely,
Yuhan Li
Reviewer 2 Report
The manuscript entitled “Excellently toughening 2,6-diaminopyridine derived poly(urethane urea) by dynamic cross-linkages and interfering hard segments from partially coordinated ligands” describes the synthesis of robust thermoplastic urethanes (TPUs). This was achieved via dynamic crosslinking through metal-ligand interactions between Zn2+ centers and pyridine moieties in the backbone of poly(urethane urea) (PUU), which was prepared using 2,6-diaminopyridine as a chain extender and ligand, while poly(propylene glycol) was employed as a soft segment. The greatest enhancement of robustness was obtained at a Zn2+/pyridine molar ratio of ¼, with enhancements in the tensile strength, elongation, and toughness of this material. The authors found that the presence of vacant pyridine ligands was a key factor in enhancing the reformation of crosslinking coordination bonds during stretching.
The authors have employed a variety of methods to characterize these materials and to evaluate their mechanical properties, including UV-visible spectroscopy, FT-IR spectroscopy, tensile tests, stress-strain analysis, and they have also proposed a viable mechanism through which toughening takes place. These characterizations were performed in a systematic manner at various ratios between the metal and the pyridine ligands. Possibly another area that could have benefited from characterization would have included microscopy observation of these materials such as transmission electron microscopy or scanning electron microscopy.
This research is of interest from both a fundamental scientific perspective as well as from an applied standpoint, as the strategy presented by the authors provides a significant enhancement in the mechanical properties of TPUs in a systematic manner. I believe that this work should be of interest to the readers of Polymers, and the researcher presented by the authors fits well within the scope of this journal. The work was conducted in a systematic matter at various Zn2+/pyridine molar ratios, and the effects of this ratio on various properties of these materials were investigated. Overall, I believe that this manuscript is worthy of publication pending minor revisions.
Overall the manuscript was reasonably well-written, but it could benefit from minor polishing, which is outlined below:
Page 1, article title: The title of this manuscript “Excellently toughening 2,6-diaminopyridine derived poly(urethane urea) by dynamic cross-linkages and interfering hard segments from partially coordinated ligands” is a little unclear and may require revision. Possibly one alternative may be “Excellent toughening of 2,6-diaminopyridine derived poly(urethane urea) via dynamic cross-linkages and interfering hard segments from partially coordinated ligands” but the phrase “and interfering hard segments from partially coordinated ligands” is still a little unclear.
Page 1, abstract, line 4: “improving tensile” can be changed to “improving the tensile”.
Page 1, abstract, Line 11: “cyclic tensile test and” can be changed to “cyclic tensile tests”.
Page 1, abstract, line 12: “significantly interferes hydrogen bonding of urea groups,” can be changed to “significantly interferes with the hydrogen bonding of urea groups,” or maybe “significantly disrupts the hydrogen bonding of urea groups,”.
Page 1, introduction, line 2: “consists of long-chain” can possibly be changed to “consists of a long-chain”.
Page 1, introduction, lines 4-7: “Conventionally, tuning mechanical properties, thermal resistance and other special features of TPU lies in altering structure of soft and hard segments because its elasticity is essentially originated from physical network constructed by micro-domains deriving from” can possibly be changed to “Conventionally, tuning the mechanical properties, thermal resistance and other special features of TPU lies in altering the structures of the soft and hard segments because its elasticity is essentially originated from the physical network constructed by micro-domains derived from”.
Page 2, introduction, 2nd paragraph, line 2: “in-situ synthesizing methods” can be changed to “in-situ synthetic methods”.
Page 2, 2nd paragraph, line5: The phrase “that enable to rupture prior to the cleavage of major structural linkage” is unclear.
Page 2, 2nd paragraph, line 9: “tremendous amount of hydrogen bonds” can possibly be changed to “a large quantity of hydrogen bonds” or maybe “the presence of numerous hydrogen bonds in”.
Page 2, 2nd paragraph, line 16: The phrase “imposes negligible on backbone mobility” is a little unclear or is missing information because it is not indicated what is imposed here. Maybe this can be changed to “imposes a negligible effect on backbone mobility”.
Page 2, 2nd paragraph, lines 16-17: “is accompanied by effect of physical”. can possibly be changed to “is accompanied by the effect of the physical”.
Page 2, 2nd paragraph, line 22: “that robustness of TPU” can possibly be changed to “that the robustness of TPU”.
Page 2, introduction, 2nd paragraph, line 26: “is still necessary to be raveled” can possibly be changed to “still requires investigation” or “remains unknown”.
Page 2, introduction, 3rd paragraph, line 4: “2,6-diaminopyridine” can be changed to “2,6-Diaminopyridine” (normally a lower case d can be used in this name, but the upper case is used here as this name is at the beginning of a sentence).
Page 2, introduction, 3rd paragraph, line 6: “2, 6-diaminopyridine” can be changed to “2,6-diaminopyridine” (with the space after the comma between 2 and 6 removed).
Page 2, introduction, 3rd paragraph, line 7: “our work focus on revealing role of” can be changed to “our work focuses on revealing the role of”.
Page 2, materials and methods, section 2.1, line 2: “4,4’-diphenylmethane diisocyanate” can be changed to “4,4’-Diphenylmethane diisocyanate”.
Page 2, materials and methods, section 2.1, line 5: “p-chlorophenol” can be changed to “p-Chlorophenol” (lower case C would normally be correct but in this case this name appears at the beginning of a sentence).
Page 3, 1st paragraph, line 8: “p-chlorophenol” can be changed to “p-Chlorophenol” (a lower case C would normally be correct but in this case this name appears at the beginning of a sentence).
Page 3, 1st paragraph, line 9: “until NCO peak disappeared in FTIR” can be changed to “until the NCO peak disappeared from the FTIR”.
Page 3, 1st paragraph, line 11: “For introducing ratio of” can possibly be changed to “To introduce a ratio of”.
Page 3, 1st paragraph, line 16: “before chain-extending with” can be changed to “prior to chain-extension with”.
Page 3, Scheme 1 caption: “Synthesizing procedure of PUU-Py elastomers” can possibly be changed to “Synthetic pathway toward the PUU-Py elastomers”.
Page 3, Section 2.2.1 Preparation of Films, line 2: “was brought into oven and” can be changed to “was placed in an oven and”.
Page 3, Section 2.2.1 Mechanical Properties Tests” This Section number is incorrect as the previous section (Preparation of Films) was also numbered as Section 2.2.1. This second section numbered 2.2.1 should be numbered 2.2.2. Also, all subsequent section numbers within Section 2.2 (Section 2.2.2 Stress relaxation tests, Section 2.2.3 Fourier transform infrared spectroscopy, and Section 2.2.4 UV-vis spectroscopy) will also be incorrect or have incorrect Section numbers (they should probably be changed to 2.2.3, 2.2.4, and 2.2.5, respectively).
Page 3, 2.2.1. Mechanical properties tests (This section should probably be renumbered as Section 2.2.2), line 3: “in a constant speed” can possibly be changed to “at a constant speed” or “at a constant rate”.
Page 4, Section 3, lines 1-2: “dangling ligand moieties along backbone was effective in enhancing” can possibly be changed to “that dangling ligand moieties along the backbone was an effective means of enhancing”.
Page 4, Section 3, lines 4-5: “might impose interference on hydrogen bonding” can be changed to “might impose interference with the hydrogen bonding” or “might disrupt the hydrogen bonding”.
Page 5, 2nd paragraph, line 1: “In present work,” can be changed to “In the present work,”.
Page 5, 2nd paragraph, line 2: “because mechanical properties” can be changed to “because the mechanical properties”.
Page 5, 2nd paragraph , line 7: “largest extend.” Can be changed to “largest extent.”.
Page 5, Section 3.2 (below Figure 2), line 4: “cm-1” should be changed to “cm-1” (with -1 written as a superscript).
Page 5, Section 3.2 (below Figure 2), line 12: “moieties is significantly affected” can be changed to “moieties are significantly affected”.
Page 5, Section 3.2 (Below Figure 2), line 13: “indicating Zn2+ ions does not” can be changed to “indicating that the Zn2+ ions do not”.
Page 6, line 1: “as content of Zn2+ increases” can possibly be changed to “as the Zn2+ content increases”.
Page 6, section 3.3, lines 4-6 (below Figure 4): “The variation of the two featured peaks versus Zn2+/pyridine ratio illustrated in Figure 3b reveals that complete complexation requires ~0.5 of Zn2+/pyridine ratio because” can possibly be changed to “The variation of the two featured peaks versus the Zn2+/pyridine ratio illustrated in Figure 3b reveals that complete complexation requires a Zn2+/pyridine molar ratio of ~0.5 because”.
Page 7, line 13: “of urea carbonyl takes place coordination with” can possibly be changed to “of the urea carbonyl undergoes coordination with” or possibly “of the urea carbonyl participates in coordination with”.
Page 7, Figure 5b: Error bars may be needed for the bar graph in Figure 5b (Hysteresis energy versus Zn2+/pyridine ratio and Residual strain versus Zn2+/pyridine ratio).
Page 7, section 3.4, line 1 (below Figure 5): “tensile test was performed” can be changed to “tensile tests were performed”.
Page 8, line 2: “could significantly interfere hydrogen bonding of urea” can possibly be changed to “could significantly interfere with the hydrogen bonding of urea” or “could significantly disrupt the hydrogen bonding of urea”.
Page 8, lines 3-4: “thus recovery of chain configuration and conformation is favored, generating smaller hysteresis loop” can be changed to “thus recovery of the chain configuration and conformation is favored, generating a smaller hysteresis loop”.
Page 8, line 5: “regardless to” can be changed to “regardless of the”.
Page 8, line 10: “with increasing of Zn2+ ions introduced” can be changed to “with the introduction of more Zn2+ ions”.
Page 8, section 3.5, line 3 (below Figure 6): “Hydrogen bonded hard segments remains intact” can possibly be changed to “The hydrogen bonded hard segments remain intact”.
Page 9, line 1: “bonds significant restrict extended chains from mutual sliding” can possibly be changed to “bonds significantly restrict extended chains from undergoing mutual sliding” or “bonds impose a significant restriction on the ability of extended chains to undergo mutual sliding”.
Page 9, section 3.6, line 6: “it displays extremely slow relaxation process, which indicates that extending of” can possibly be changed to “an extremely slow relaxation process is observed, which indicates that the extension of”
Page 10, line 5: “which acts as physical cross-linkage to” can possibly be changed to “which act as physical cross-linking junctions to”.
Page 10, line 3 (below Scheme 1): “has the most excellent” can be changed to “has the best”.
Page 10, line 5 (below Scheme 1): The phrase “does not further favor in enhancing mechanical properties any more” is a little unclear. Possibly this can be changed to “does not yield further enhancements of the mechanical properties”.
Page 10, lines 6-8 (below Scheme 1): “demonstrates the optimal ratio is not necessarily corresponding to complexation of entire ligands, which implies that remaining vacant ligands for reformation of noncovalent cross-linkages during stretching is the key for improving robustness” can possibly be changed to “demonstrates that the optimal ratio does not necessarily correspond to the complexation of every ligand, which implies that the present of vacant ligands facilitates the reformation of noncovalent cross-linkages during stretching and is the key for improving robustness”.
Page 10, lines 10-11 (below Scheme 1): “for external force to destruct hard segments derived from hydrogen bonding due to its relatively weak” can possibly be changed to “for external forces to destruct hard segments derived from hydrogen bonding interactions due to their relatively weak”.
Page 10, line 14 (below Scheme 1): “less or no” can be changed to “fewer or no”.
Page 10, line 19 (below Scheme 1): “rest half of vacant ligands” can possibly be changed to “remaining half of vacant ligands”.
Page 10, conclusions, line 1: “In summary, incorporating pyridine moiety into backbone of polypropylene glycol based” can be changed to “In summary, the incorporation of pyridine moieties into the backbone of polypropylene glycol-based”.
Page 10, conclusions, line 2: “improve robustness upon complexed with 1/4 equivalent” can possibly be changed to “improve the robustness of this material upon complexation with 1/4 equivalents of”.
Page 11, lines 4-5: “ligands is found to be the most effective for enhancing robustness because the rest half of ligands are” can possibly be changed to “ligands is found to provide the most effective robustness enhancement because the remaining half of the ligands are”.
Page 11, line 7: “meaningful for the industrial application” can possibly be changed to “meaningful for industrial applications”.
Author Response
Dear reviewers:
We thank you for your careful consideration and thoughtful comments on our manuscript (ID: polymers-561361). We have carefully read your comments and suggestions which are valuable and helpful for improving our manuscript. We have make corrections basically as you suggested and all the changes are marked in red in the revised manuscript. The corrections are also detailed as follows:
Point 1: Page 1, article title: The title of this manuscript “Excellently toughening 2,6-diaminopyridine derived poly(urethane urea) by dynamic cross-linkages and interfering hard segments from partially coordinated ligands” is a little unclear and may require revision. Possibly one alternative may be “Excellent toughening of 2,6-diaminopyridine derived poly(urethane urea) via dynamic cross-linkages and interfering hard segments from partially coordinated ligands” but the phrase “and interfering hard segments from partially coordinated ligands” is still a little unclear. 

Response 1: The title is revised as “Excellent toughening of 2,6-diaminopyridine derived poly(urethane urea) via dynamic cross-linkages and interfering with hydrogen bonding of urea groups from partially coordinated ligands”. The interference to hard segments is actually affecting hydrogen bonding of urea groups.
Point 2: Page 1, abstract, line 4: “improving tensile” can be changed to “improving the tensile”.
Response 2: The phrase has been replaced as the reviewer suggested in the manuscript.
Point 3: Page 1, abstract, Line 11: “cyclic tensile test and” can be changed to “cyclic tensile tests”.
Response 3: The phrase has been replaced as the reviewer suggested in the manuscript.
Point 4: Page 1, abstract, line 12: “significantly interferes hydrogen bonding of urea groups,” can be changed to “significantly interferes with the hydrogen bonding of urea groups,” or maybe “significantly disrupts the hydrogen bonding of urea groups”.
Response 4: The phrase has been replaced with “significantly interferes with the hydrogen bonding of urea groups” in the manuscript.
Point 5: Page 1, introduction, line 2: “consists of long-chain” can possibly be changed to “consists of a long-chain”.
Response 5: The phrase has been replaced as the reviewer suggested in the manuscript.Point 6: Page 1, introduction, lines 4-7: “Conventionally, tuning mechanical properties, thermal resistance and other special features of TPU lies in altering structure of soft and hard segments because its elasticity is essentially originated from physical network constructed by micro-domains deriving from” can possibly be changed to “Conventionally, tuning the mechanical properties, thermal resistance and other special features of TPU lies in altering the structures of the soft and hard segments because its elasticity is essentially originated from the physical network constructed by micro-domains derived from”.
Point 6: Page 1, introduction, lines 4-7: “Conventionally, tuning mechanical properties, thermal resistance and other special features of TPU lies in altering structure of soft and hard segments because its elasticity is essentially originated from physical network constructed by micro-domains deriving from” can possibly be changed to “Conventionally, tuning the mechanical properties, thermal resistance and other special features of TPU lies in altering the structures of the soft and hard segments because its elasticity is essentially originated from the physical network constructed by micro-domains derived from”.
Response 6: The phrase has been replaced as the reviewer suggested in the manuscript.
Point 7: Page 2, introduction, 2nd paragraph, line 2: “in-situ synthesizing methods” can be changed to “in-situ synthetic methods”..
Response 7: The phrase has been replaced as the reviewer suggested in the manuscript.
Point 8: Page 2, 2nd paragraph, line5: The phrase “that enable to rupture prior to the cleavage of major structural linkage” is unclear.
Response 8: Please provide your response for Point 2. (in red)This phrase could be revised as “which perform rupture of their binding prior to the cleavage of major structural linkage upon applied force”
Point 9: Page 2, 2nd paragraph, line 9: “tremendous amount of hydrogen bonds” can possibly be changed to “a large quantity of hydrogen bonds” or maybe “the presence of numerous hydrogen bonds in”.
Response 9: The phrase has been replaced with “the presence of numerous hydrogen bonds in” in the manuscript.
Point 10: Page 2, 2nd paragraph, line 16: The phrase “imposes negligible on backbone mobility” is a little unclear or is missing information because it is not indicated what is imposed here. Maybe this can be changed to “imposes a negligible effect on backbone mobility”.
Response 10: The phrase has been replaced as the reviewer suggested in the manuscript.
Point 11: Page 2, 2nd paragraph, lines 16-17: “is accompanied by effect of physical”. can possibly be changed to “is accompanied by the effect of the physical”.
Response 11: The phrase has been replaced as the reviewer suggested in the manuscript.
Point 12: Page 2, 2nd paragraph, line 22: “that robustness of TPU” can possibly be changed to “that the robustness of TPU”.
Response 12: The phrase has been replaced as the reviewer suggested in the manuscript.
Point 13: Page 2, introduction, 2nd paragraph, line 26: “is still necessary to be raveled” can possibly be changed to “still requires investigation” or “remains unknown”.
Response 13: The phrase has been replaced with “still requires investigation” in the manuscript.
Point 14: Page 2, introduction, 3rd paragraph, line 4: “2,6-diaminopyridine” can be changed to “2,6-Diaminopyridine” (normally a lower case d can be used in this name, but the upper case is used here as this name is at the beginning of a sentence).
Response 14: The phrase has been replaced as the reviewer suggested in the manuscript.
Point 15: Page 2, introduction, 3rd paragraph, line 6: “2, 6-diaminopyridine” can be changed to “2,6-diaminopyridine” (with the space after the comma between 2 and 6 removed).
Response 15: The phrase has been replaced as the reviewer suggested throughout the entire manuscript.
Point 16: Page 2, introduction, 3rd paragraph, line 7: “our work focus on revealing role of” can be changed to “our work focuses on revealing the role of”.
Response 16: The phrase has been replaced as the reviewer suggested in the manuscript.
Point 17: Page 2, materials and methods, section 2.1, line 2: “4,4’-diphenylmethane diisocyanate” can be changed to “4,4’-Diphenylmethane diisocyanate”.
Response 17: The phrase has been replaced as the reviewer suggested in the manuscript.
Point 18: Page 2, materials and methods, section 2.1, line 5: “p-chlorophenol” can be changed to “p-Chlorophenol” (lower case C would normally be correct but in this case this name appears at the beginning of a sentence).
Response 18: The phrase has been replaced as the reviewer suggested in the manuscript.
Point 19: Page 3, 1st paragraph, line 8: “p-chlorophenol” can be changed to “p-Chlorophenol” (a lower case C would normally be correct but in this case this name appears at the beginning of a sentence).
Response 19: The phrase has been replaced as the reviewer suggested in the manuscript.
Point 20: Page 3, 1st paragraph, line 9: “until NCO peak disappeared in FTIR” can be changed to “until the NCO peak disappeared from the FTIR”.
Response 20: The phrase has been replaced as the reviewer suggested in the manuscript.
Point 21: Page 3, 1st paragraph, line 11: “For introducing ratio of” can possibly be changed to “To introduce a ratio of”.
Response 21: The phrase has been replaced as the reviewer suggested in the manuscript.
Point 22: Page 3, 1st paragraph, line 16: “before chain-extending with” can be changed to “prior to chain-extension with”.
Response 22: The phrase has been replaced as the reviewer suggested in the manuscript.
Point 23: Page 3, Scheme 1 caption: “Synthesizing procedure of PUU-Py elastomers” can possibly be changed to “Synthetic pathway toward the PUU-Py elastomers”.
Response 23: The phrase has been replaced as the reviewer suggested in the manuscript.
Point 24: Page 3, Section 2.2.1 Preparation of Films, line 2: “was brought into oven and” can be changed to “was placed in an oven and”.
Response 24: The phrase has been replaced as the reviewer suggested in the manuscript.
Point 25: Page 3, Section 2.2.1 Mechanical Properties Tests” This Section number is incorrect as the previous section (Preparation of Films) was also numbered as Section 2.2.1. This second section numbered 2.2.1 should be numbered 2.2.2. Also, all subsequent section numbers within Section 2.2 (Section 2.2.2 Stress relaxation tests, Section 2.2.3 Fourier transform infrared spectroscopy, and Section 2.2.4 UV-vis spectroscopy) will also be incorrect or have incorrect Section numbers (they should probably be changed to 2.2.3, 2.2.4, and 2.2.5, respectively)..
Response 25: All subsequent section numbers have been corrected in the manuscript.
Point 26: Page 3, 2.2.1. Mechanical properties tests (This section should probably be renumbered as Section 2.2.2), line 3: “in a constant speed” can possibly be changed to “at a constant speed” or “at a constant rate”.
Response 26: The phrase has been replaced as the reviewer suggested in the manuscript.
Point 27: Page 4, Section 3, lines 1-2: “dangling ligand moieties along backbone was effective in enhancing” can possibly be changed to “that dangling ligand moieties along the backbone was an effective means of enhancing”.
Response 27: The phrase has been replaced as the reviewer suggested in the manuscript.
Point 28: Page 4, Section 3, lines 4-5: “might impose interference on hydrogen bonding” can be changed to “might impose interference with the hydrogen bonding” or “might disrupt the hydrogen bonding”.
Response 28: The phrase has been replaced as the reviewer suggested in the manuscript.
Point 29: Page 5, 2nd paragraph, line 1: “In present work,” can be changed to “In the present work,”.
Response 29: The phrase has been replaced as the reviewer suggested in the manuscript.
Point 30: Page 5, 2nd paragraph, line 2: “because mechanical properties” can be changed to “because the mechanical properties”.
Response 30: The phrase has been replaced as the reviewer suggested in the manuscript.
Point 31: Page 5, 2nd paragraph, line 7: “largest extend.” Can be changed to “largest extent.”.
Response 31: The word has been replaced as the reviewer suggested in the manuscript.
Point 32: Page 5, Section 3.2 (below Figure 2), line 4: “cm-1” should be changed to “cm-1” (with -1 written as a superscript)..
Response 32: The phrase has been replaced as the reviewer suggested in the manuscript.
Point 33: Page 5, Section 3.2 (below Figure 2), line 12: “moieties is significantly affected” can be changed to “moieties are significantly affected”.
Response 33: The phrase has been replaced as the reviewer suggested in the manuscript.
Point 34: Page 5, Section 3.2 (Below Figure 2), line 13: “indicating Zn2+ ions does not” can be changed to “indicating that the Zn2+ ions do not”.
Response 34: The phrase has been replaced as the reviewer suggested in the manuscript.
Point 35: Page 6, line 1: “as content of Zn2+ increases” can possibly be changed to “as the Zn2+ content increases”.
Response 35: The phrase has been replaced as the reviewer suggested in the manuscript.
Point 36: Page 6, section 3.3, lines 4-6 (below Figure 4): “The variation of the two featured peaks versus Zn2+/pyridine ratio illustrated in Figure 3b reveals that complete complexation requires ~0.5 of Zn2+/pyridine ratio because” can possibly be changed to “The variation of the two featured peaks versus the Zn2+/pyridine ratio illustrated in Figure 3b reveals that complete complexation requires a Zn2+/pyridine molar ratio of ~0.5 because”
Response 36: The phrase has been replaced as the reviewer suggested in the manuscript. Similar incorrect expression in other sentences of the context has also been revised.
Point 37: Page 7, line 13: “of urea carbonyl takes place coordination with” can possibly be changed to “of the urea carbonyl undergoes coordination with” or possibly “of the urea carbonyl participates in coordination with”
Response 37: The phrase has been replaced as the reviewer suggested in the manuscript.
Point 38: Page 7, Figure 5b: Error bars may be needed for the bar graph in Figure 5b (Hysteresis energy versus Zn2+/pyridine ratio and Residual strain versus Zn2+/pyridine ratio).
Response 38: The error bars were accidentally missed during the data processing via Origin. Now, a newly-plotted Figure 5b with error bars is presented in the manuscript.
Point 39: Page 7, section 3.4, line 1 (below Figure 5): “tensile test was performed” can be changed to “tensile tests were performed”
Response 39: The phrase has been replaced as the reviewer suggested in the manuscript.
Point 40: Page 8, line 2: “could significantly interfere hydrogen bonding of urea” can possibly be changed to “could significantly interfere with the hydrogen bonding of urea” or “could significantly disrupt the hydrogen bonding of urea”
Response 40: The phrase has been replaced as the reviewer suggested in the manuscript.
Point 41: Page 8, lines 3-4: “thus recovery of chain configuration and conformation is favored, generating smaller hysteresis loop” can be changed to “thus recovery of the chain configuration and conformation is favored, generating a smaller hysteresis loop”
Response 41: The phrase has been replaced as the reviewer suggested in the manuscript.
Point 42: Page 8, line 5: “regardless to” can be changed to “regardless of the”.
Response 42: The phrase has been replaced as the reviewer suggested in the manuscript.
Point 43: Page 8, line 10: “with increasing of Zn2+ ions introduced” can be changed to “with the introduction of more Zn2+ ions”
Response 43: The phrase has been replaced as the reviewer suggested in the manuscript.
Point 44: Page 8, section 3.5, line 3 (below Figure 6): “Hydrogen bonded hard segments remains intact” can possibly be changed to “The hydrogen bonded hard segments remain intact”
Response 44: The phrase has been replaced as the reviewer suggested in the manuscript.
Point 45: Page 9, line 1: “bonds significant restrict extended chains from mutual sliding” can possibly be changed to “bonds significantly restrict extended chains from undergoing mutual sliding” or “bonds impose a significant restriction on the ability of extended chains to undergo mutual sliding”
Response 45: The phrase has been replaced as the reviewer suggested in the manuscript.
Point 46: Page 9, section 3.6, line 6: “it displays extremely slow relaxation process, which indicates that extending of” can possibly be changed to “an extremely slow relaxation process is observed, which indicates that the extension of
Response 46: The phrase has been replaced as the reviewer suggested in the manuscript.
Point 47: Page 10, line 5: “which acts as physical cross-linkage to” can possibly be changed to “which act as physical cross-linking junctions to”
Response 47: The phrase has been replaced as the reviewer suggested in the manuscript.
Point 48: Page 10, line 3 (below Scheme 1): “has the most excellent” can be changed to “has the best”.
Response 48: The phrase has been replaced as the reviewer suggested in the manuscript.
Point 49: Page 10, line 5 (below Scheme 1): The phrase “does not further favor in enhancing mechanical properties any more” is a little unclear. Possibly this can be changed to “does not yield further enhancements of the mechanical properties”.
Response 49: The phrase has been replaced as the reviewer suggested in the manuscript.
Point 50: Page 10, lines 6-8 (below Scheme 1): “demonstrates the optimal ratio is not necessarily corresponding to complexation of entire ligands, which implies that remaining vacant ligands for reformation of noncovalent cross-linkages during stretching is the key for improving robustness” can possibly be changed to “demonstrates that the optimal ratio does not necessarily correspond to the complexation of every ligand, which implies that the present of vacant ligands facilitates the reformation of noncovalent cross-linkages during stretching and is the key for improving robustness”.
Response 50: The phrase has been replaced as the reviewer suggested in the manuscript.
Point 51: Page 10, lines 10-11 (below Scheme 1): “for external force to destruct hard segments derived from hydrogen bonding due to its relatively weak” can possibly be changed to “for external forces to destruct hard segments derived from hydrogen bonding interactions due to their relatively weak”.
Response 51: The phrase has been replaced as the reviewer suggested in the manuscript.
Point 52: Page 10, line 14 (below Scheme 1): “less or no” can be changed to “fewer or no”.
Response 52: The phrase has been replaced as the reviewer suggested in the manuscript.
Point 53: Page 10, line 19 (below Scheme 1): “rest half of vacant ligands” can possibly be changed to “remaining half of vacant ligands”.
Response 53: The phrase has been replaced as the reviewer suggested in the manuscript.
Point 54: Page 10, conclusions, line 1: “In summary, incorporating pyridine moiety into backbone of polypropylene glycol based” can be changed to “In summary, the incorporation of pyridine moieties into the backbone of polypropylene glycol-based”.
Response 54: The phrase has been replaced as the reviewer suggested in the manuscript.
Point 55: Page 10, conclusions, line 2: “improve robustness upon complexed with 1/4 equivalent” can possibly be changed to “improve the robustness of this material upon complexation with 1/4 equivalents of”
Response 55: The phrase has been replaced as the reviewer suggested in the manuscript.
Point 56: Page 11, lines 4-5: “ligands is found to be the most effective for enhancing robustness because the rest half of ligands are” can possibly be changed to “ligands is found to provide the most effective robustness enhancement because the remaining half of the ligands are”.
Response 56: The phrase has been replaced as the reviewer suggested in the manuscript.
Point 57: “meaningful for the industrial application” can possibly be changed to “meaningful for industrial applications.
Response 57: The phrase has been replaced as the reviewer suggested in the manuscript.
